# Neuronal Pentraxin 2 as a Potential Biomarker for Nusinersen Therapy Response in Adults with Spinal Muscular Atrophy: A Pilot Study

**DOI:** 10.3390/biomedicines13081821

**Published:** 2025-07-25

**Authors:** Svenja Neuhoff, Linda-Isabell Schmitt, Kai Christine Liebig, Stefanie Hezel, Nick Isana Tilahun, Christoph Kleinschnitz, Markus Leo, Tim Hagenacker

**Affiliations:** Department of Neurology and Center for Translational Neuro- and Behavioral Sciences (C-TNBS), University Hospital Essen, Hufelandstr. 55, 45147 Essen, Germany; svenja.neuhoff@uk-essen.de (S.N.); linda-isabell.schmitt@uk-essen.de (L.-I.S.); kaichristine.liebig@uk-essen.de (K.C.L.); stefanie.hezel@uk-essen.de (S.H.); nick.tilahun@web.de (N.I.T.); christoph.kleinschnitz@uk-essen.de (C.K.); markus.leo@uk-essen.de (M.L.)

**Keywords:** spinal muscular atrophy, biomarker, NPTX2, nusinersen

## Abstract

**Background:** The treatment landscape for spinal muscular atrophy (SMA) has changed significantly with the approval of gene-based therapies such as nusinersen for adults with SMA (pwSMA). Despite their efficacy, high costs and treatment burden highlight the need for biomarkers to objectify or predict treatment response. This study aimed to identify such biomarkers. **Methods:** A proteomic analysis of cerebrospinal fluid (CSF) from pwSMA (*n* = 7), who either significantly improved (SMA Improvers) or did not improve in motor function (SMA Non-Improvers) under nusinersen therapy, was performed. Data are available via ProteomeXchange with identifier PXD065345. Candidate biomarkers—Neuronal Pentraxin 2 (NPTX2), Contactin 5 (CNTN5), and Anthrax Toxin Receptor 1 (ANTXR1)—were investigated by ELISA in serum and CSF from an independent pwSMA cohort (*n* = 14) at baseline, 2 and 14 months after therapy initiation. Biomarker concentrations were correlated with clinical outcomes. Additionally, NPTX2 was stained in spinal cord sections from a mild SMA mouse model (FVB.Cg-Smn1tm1Hung Tg(SMN2)2Hung/J). **Results:** CSF NPTX2 levels decreased in pwSMA after 14 months of nusinersen therapy, independent of clinical response. The change in NPTX2 serum levels over 14 months of nusinersen treatment correlated with the change in HFMSE during this period. CNTN5 and ANTXR1 showed no significant changes. In the SMA mouse model, NPTX2 immunoreactivity increased at motoneuron loss onset. **Conclusions:** NPTX2 emerges as a potential biomarker of treatment response to nusinersen in pwSMA suggesting its significant pathophysiological role in late-onset SMA, warranting further investigation.

## 1. Introduction

5q-associated Spinal Muscular Atrophy (SMA) is a hereditary motor neuron disease leading to progressive weakness and atrophy of limb, bulbar, and respiratory muscles caused by a homozygous deletion or compound heterozygosity with deletion and point mutation in the survival motor neuron 1 (*SMN1*) gene [1]. Survival motor neuron (SMN) protein deficiency causes progressive motoneuronal degeneration in the anterior horn of the spinal cord and can only be insufficiently compensated by truncated SMN protein deriving from the survival motor neuron 2 (*SMN2*) gene of variable copy number [2,3]. The disease classification ranges from type 1, characterized by a low *SMN2* copy number and a naturally early-onset, severe disease course with a shortened life expectancy, to type 4, featuring a higher *SMN2* copy number, later disease onset in adulthood, and normal life expectancy [4,5].

Recently, disease-modifying therapies (nusinersen, risdiplam, and onasemnogene abeparvovec) have been approved for treatment of SMA, with nusinersen and risdiplam showing improvement of motor function even in adults [6,7,8,9]. While these treatments are proven effective, therapy costs and efforts are high. Treatment decisions are currently not based on clear evidence-based criteria but on patient preference, unless, for example, technical conditions such as severe scoliosis or spondylodesis prohibit the intrathecal administration of nusinersen.

Considering expanding therapeutic options for SMA, it is essential to establish objective decision criteria for the selection of specific therapies. In this regard, the utilization of biomarkers could serve to objectively measure treatment success and predict treatment response. Although clinical, structural-morphological, and functional assessments are valuable, they are often challenging to assess, rater-dependent, and limited in their sensitivity [10,11]. Blood and cerebrospinal fluid (CSF) biomarkers are easier to obtain, quantitative, and can be evaluated objectively. Neurofilaments are becoming established blood and CSF biomarkers for Amyotrophic Lateral Sclerosis and Multiple Sclerosis [12,13]. They have also been investigated in SMA [14,15,16]. A recent large-scale study investigating neurofilament light chain (NfL) levels longitudinally in 113 adults with SMA under nusinersen therapy found higher NfL levels in the SMA than in the control group at baseline, but no predictive value of NfL regarding clinical parameters [15]. A moderate decrease in NfL concentration in CSF was observed under nusinersen treatment, but no correlation with clinical parameters was found [15].

Proteomics enables the comprehensive analysis of proteins in biological samples, allowing for the detection of changes in protein expression or modification [17]. These changes can serve as potential biomarkers indicating treatment response at the molecular or even clinical level. To identify useful biomarkers for adult SMA, we quantified candidate biomarkers identified through CSF-based proteome analysis in adults with SMA who showed different responses to nusinersen therapy.

## 2. Materials and Methods

### 2.1. Study Design

This translational study employed a multi-step approach to identify and validate potential biomarkers in the CSF and serum of adults with SMA undergoing nusinersen therapy. The study was conducted in the Department of Neurology, University Hospital, Essen, Germany. Participants provided written informed consent prior to their inclusion in the study. The study was approved by the local ethics committee of the University of Duisburg-Essen, Germany (approval number 18-8071-BO, approve date 29 May 2018).

We included adult individuals with molecularly confirmed (homozygous deletion of *SMN1*) 5q-associated SMA, referred to as persons with SMA (pwSMA), under treatment with nusinersen as well as healthy non-SMA individuals as controls. Controls were between 28 and 57 years old and underwent lumbar puncture as part of diagnostic work-up for headache or transient sensory disturbances. Inflammatory, degenerative, ischemic, neoplastic or metabolic diseases of the central nervous system were ruled out in controls. Assessments and samples of pwSMA (serum and CSF) were collected within the regular treatment schedule for administration of nusinersen between 2017 and 2021.

The study consisted of three main methodological phases (see Figure 1). First, an unbiased proteomic analysis of CSF was conducted in a small cohort of seven pwSMA with different clinical responses to nusinersen therapy, as well as from two controls. PwSMA were classified as “SMA Improvers” if they showed a ≥3-point increase in the Hammersmith Functional Motor Scale Expanded (HFMSE) Score after 6 months of treatment and as “SMA Non-Improvers” if they showed less or no improvement.

In a second, hypothesis-driven approach, selected proteins were analyzed through quantification by ELISA in the CSF and serum in an independent sample of fourteen pwSMA with different clinical responses to nusinersen and four controls. Assessments were carried out at multiple time points: baseline (T0), 2 months (T1), and 14 months (T2), with clinical follow-up at baseline, 2, 6, and 14 months.

To further validate the biological significance of the CSF and serum findings in pwSMA, a small-scale analysis of spinal cord tissue from a late-onset SMA mouse model was conducted. Three SMA mice and three wild-type (wt) mice were analyzed at postnatal days P20, P42, P70, and >P400. An overview of the analyses performed and the samples included is provided in Figure 1. In the following sections, each step of the study is described in more detail.

### 2.2. Assessment of Motor Function

We used the HFMSE and Revised Upper Limb Module (RULM) to assess motor function. Tests were conducted within regular visits at baseline, 2 months, 6 months, and 14 months after treatment initiation with nusinersen. HFMSE and RULM are both established and validated tools for evaluating motor function and disease progression in individuals with SMA types 2 and 3 [18,19]. The HFMSE assesses motor function of the extremities, trunk, and head with 33 items scored from 0 to 2, resulting in a maximum score of 66. Higher scores indicate better motor function. The RULM evaluates upper extremity function using 19 items. One item is scored 0 or 1, and the remaining 18 are scored 0 to 2 points resulting in a maximum score of 37 with higher scores indicating better motor function.

### 2.3. Proteome Analysis from CSF

Unbiased proteomic profiling was performed on CSF samples from seven pwSMA and two age-matched controls. Of the seven pwSMA included in the proteomic analysis, five were Improvers and two Non-Improvers. Given the exploratory nature of this analysis, the unbalanced sample was chosen based on availability and the expectation of greater biological dynamics in a subgroup with pronounced clinical change. Demographic and clinical characteristics of pwSMA are summarized in Table 1. Sample composition and measurement time points (baseline and 6 months) are depicted in Figure 1.

For each sample, 50 µL of CSF were reduced, alkylated, and digested overnight with trypsin. The resulting peptides were desalted using C18 spin columns, dried, and reconstituted in LC solvent containing indexed retention time (iRT) peptides for calibration. Peptide concentrations were quantified via BCA assay. Equal peptide volumes per group and time point were pooled and fractionated by high-pH reversed-phase chromatography (HPRP) into six fractions. LC-MS/MS analysis was performed on a Thermo Scientific™ Q Exactive™ HF (Thermo Fisher Scientific, Bremen, Germany) mass spectrometer coupled to an Easy-nLC 1200 nano-LC system (Thermo Fisher Scientific, San Jose, CA, USA). For spectral library generation, pooled study samples and commercial CSF were analyzed using data-dependent acquisition (DDA). Data-independent acquisition (DIA, HRM) with 22 windows per sample was used for quantification. All raw data were processed using Spectronaut Pulsar version 13.0. (Biognosys, Schlieren, Switzerland), employing a hybrid spectral library generated from DDA and DIA data. Protein and peptide identifications were filtered at a 1% false discovery rate (FDR). Data normalization was performed using local regression.

Differentially expressed proteins were defined by a *p*-value < 0.05 and an average fold change > 1.5. Further statistical analysis and data visualization were conducted in R 3.6.2. Hierarchical clustering was based on Manhattan distance and Ward’s linkage. Principal component analysis (PCA) was performed using prcomp, and partial least squares discriminant analysis (PLS-DA) using the mixOmics package.

The mass spectrometry proteomics data have been deposited to the ProteomeXchange Consortium via the PRIDE partner repository with the dataset identifier PXD065345. 

### 2.4. Sample Selection for Enzyme-Linked Immunosorbent Assay (ELISA)

To reduce the risk of sampling bias and to ensure a broad spectrum of treatment responses, an independent sample was created. From a total of 28 pwSMA with available baseline CSF or serum samples, 14 were selected to represent distinct ranges of change in HFMSE score (Δ6 HFMSE) after 6 months of nusinersen treatment, covering the full observed response range (Δ6 HFMSE −6 to +20). One individual per response level was randomly selected to ensure maximal variability in treatment outcomes, rather than representativeness of the cohort. Seven of the fourteen pwSMA were classified as either Improvers or Non-Improvers. HFMSE and RULM scores were available for all pwSMA at all time points. Demographic and clinical characteristics of all included pwSMA are given in Table 1. Four non-SMA individuals served as controls.

### 2.5. ELISA from CSF and Serum

In a hypothesis-driven approach, single proteins were selected for quantification by ELISA in the cohort described in Section 2.4. Based on the results of the proteome analysis and the function of these proteins, Contactin 5 (CNTN5), mediating cell surface interactions during the development of the nervous system, and Anthrax Toxin Receptor 1 (ANTXR1), a transmembrane protein regulating cell attachment and migration, were further investigated. As the proteome analysis also revealed regulatory differences of proteins of the neuronal pentraxin family, Neuronal Pentraxin 2 (NPTX2) was additionally selected for further investigation. This decision was supported by the previous literature reports describing NPTX2 as a biomarker candidate for synaptic impairment in neurodegenerative diseases with altered expression reported in conditions such as Alzheimer’s disease and amyotrophic lateral sclerosis [20,21].

The levels of NPTX2, CNTN5, and ANTXR1 were measured in CSF and serum at three time points: baseline (T0), 2 months (T1), and 14 months (T2) after starting treatment with nusinersen. Measuring time points and types and numbers of samples that were analyzed are shown in Figure 1. To determine the levels of NPTX2, CNTN5, and ANTXR1 in CSF and serum samples, ELISA assays were performed according to the manufacturer’s protocol (#MBS8802191, #MBS1606034, #MBS941904; MyBioSource, San Diego, CA, USA).

### 2.6. Animals

The SMN-deficient mouse model FVB.Cg-Smn1tm1Hung Tg(SMN2)2Hung/J (Jackson #005058) reflects a milder SMA phenotype. The animals exhibit reduced weight after 20 days and decreased grip strength after 33 days, while maintaining a normal life expectancy. The model was chosen to reflect the disease course of milder affected pwSMA [22]. This mouse model was obtained from Jackson Laboratory and bred at the University Hospital Essen. It is homozygous for the murine *SMN1* knockout and carries four copies of human *SMN2*. Mice were maintained on a 12/12-h light/dark cycle with free access to water and standard food pellets. Animals were monitored weekly to examine body condition, weight, and general health. The experiments were conducted under the animal welfare guidelines of the University Duisburg Essen. The use of the SMA mouse model was approved by the State Agency for Nature, Environment and Consumer Protection (LANUV) in North Rhine-Westphalia (reference number 81–02.04.2020.A335, approve date 6 January 2021).

### 2.7. NPTX2 Immunostaining of Murine Spinal Cord Tissue Slices

Spinal cord tissue was collected from three mice of the SMA mouse model, referred to as “SMA mice”, each at P20 (postnatal day 20), P42, P70, and P>400. The same number of age-matched FVB/N wild type (wt) mice served as controls.

Lumbar spinal cord tissue was frozen in liquid nitrogen and stored at −80 °C before cryosections of 20 µm were prepared, with every fifth section placed on a separate microscopy slide. The tissue was fixed in 4% paraformaldehyde (PFA in PBS, 15 min), washed, permeabilized (PBS, 0.1 v/w Triton X-100, 15 min), and blocked (PBS, 5% bovine serum albumin, 1 h). Primary antibodies for motor neurons (anti-SMI-32, mouse, 1:500, #801701, BioLegends, San Diego, CA, USA) and NPTX2 (anti-NPTX2, rabbit, 1:250, #NBP2-19572, Novus Biologicals, Centennial, CO, USA) were diluted in blocking solution and incubated at 4 °C overnight. Sections were washed and secondary antibodies (goat anti-rabbit, goat anti-mouse, 1:300, Dianova, Hamburg, Germany) and DAPI (1:1000, Sigma-Aldrich, Taufkirchen, Germany) were diluted in blocking solution and incubated at room temperature.

Three slices per animal were analyzed. Images were captured with a Zeiss Axio Observer.Z1 Apotome fluorescence microscope (Zeiss, Oberkochen, Germany) and Zeiss Zensoftware version 3.11. Immunoreactivity was quantified using ImageJ software (NIH) version 1.54. Immunoreactive cells were selected using the free-hand tool, and the fluorescence intensity of NPTX2 was measured and normalized to the background in each image. Fluorescence intensity in SMA mice tissue was normalized to that of wt tissue. Following immunostaining with the motoneuronal marker SMI-32, motoneurons were manually counted.

### 2.8. Statistical Analysis

Statistical analyses were performed on the results of protein quantification by ELISA and clinical scores. Additionally, NPTX2 immunostaining and motoneuron count in spinal cord tissue of the SMA mice were analyzed.

To analyze longitudinal changes in biomarker levels across the three time points (T0, T1, T2), a Friedman test was applied. Post hoc pairwise comparisons were conducted using Wilcoxon signed-rank tests. The resulting *p*-values were adjusted for multiple comparisons using Bonferroni correction. The Mann–Whitney U test was used to compare SMA Improvers and SMA Non-Improvers. Correlations between the protein levels and motor scores HFMSE and RULM were calculated using Spearman’s rank correlation coefficient. Calculations were performed using blank-subtracted values, which were adjusted by subtracting the absorbance value of the blank (a sample without the target analyte) from the actual sample value. This helps eliminate background noise and nonspecific signals, ensuring more accurate results. The absorbance values of pwSMA were normalized to those of the controls.

To compare NPTX2 immunostaining fluorescence intensity and motoneuron number between SMA and wt mice, values from SMA spinal cord sections were normalized to the wt values on the same slide (averaged and set to 1). Each slide contained three sections from SMA and three from wt mice. At each time point (P20, P42, P70, P>400), six animals (three SMA and three wt) were examined, with up to three spinal cord sections per animal. For statistical analysis, up to nine section-level data points per time point were included across the three SMA animals. Damaged or incomplete sections were excluded. To determine whether normalized SMA values differed significantly from wt, a one-sample Wilcoxon signed-rank test was performed for each time point.

## 3. Results

### 3.1. Proteomic Profiling of CSF

A total of 117 proteins were found to be differentially regulated (*p*-value < 0.05 and average fold change > 1.5) between pwSMA and controls. Figure 2 shows the differences in protein expression between pwSMA and controls. Seventeen proteins (ANTXR1, CNTN5, CSF2RA, C1QTNF5, FUCA2, HAMP, IGHG3, IGHV3-43D, IGHV3-74, IGHV6-1, IGLC7, LARGE1, MT3, MYOC, PLOD3, RTN1, SERPINE2) were differentially regulated between SMA Improvers and SMA Non-Improvers at baseline. Between baseline and 6 months after treatment initiation, four proteins (DNS2A_HUMAN, FSTL4, NPTN, PLA2G15) were differentially regulated in SMA Improvers and three (CACHD1, DDAH1, HEXB) in SMA Non-Improvers. Among the proteins differentially regulated between SMA Improvers and SMA Non-Improvers at baseline were CNTN5 (ratio 0.21; log ratio 2.24; *p*-ANOVA = 0.01) and ANTXR1 (ratio 1.91; log ratio 0.93; p-ANOVA = 0.05) (Figure 2). NPTX2 levels were not different between SMA Improvers and SMA Non-Improvers at baseline (ratio 1.21; log ratio 0.28; *p*-ANOVA = 0.48). Regarding other proteins of the NPTX family, NPTX1 was lower at 6 months than at baseline, measured across all pwSMA (ratio 1.3; log ratio 0.38; *p*-ANOVA = 0.03). All protein ratios between the groups are available as a Appendix A.

### 3.2. ELISA-Based Quantification of NPTX2, CNTN5, and ANXTR1 in CSF and Serum

In CSF, the NPTX2 concentration showed no differences between pwSMA and controls (Figure 3a). Across all pwSMA, the NPTX2 concentration in CSF decreased over time ((χ^2^(2) = 10.571, *p* = 0.005, *n* = 7). Levels were lower at T2 compared to both T0 (z = 2.940, *p* = 0.010, *n* = 9) and T1 (z = 2.673, *p* = 0.023, *n* = 7) while no difference was observed between T0 and T1 (Figure 3a). There were no differences between SMA Improvers and SMA Non-Improvers at T0, T1, or T2 (Figure 3b) and no correlations of change in the NPTX2 CSF level and HFMSE or RULM scores from T0 to T2.

In serum, the NPTX2 concentration showed no differences between pwSMA and controls (Figure 3d). The NPTX2 serum level was higher at T2 in SMA Non-Improvers than in SMA Improvers (z = −2.030, *p* = 0.048, *n* = 12) (Figure 3e). It increased in all five SMA Non-Improvers and decreased in five of six SMA Improvers from T0 to T2, among all individuals with available data at both T0 and T2. Changes in the NPTX2 serum level from T0 to T2 correlated negatively with changes in the HFMSE from T0 to T2 (r = −0.688, *p* = 0.019, *n* = 11) across all pwSMA (Figure 4). No differences were observed between SMA Improvers and SMA Non-Improvers at T0 or T1 (Figure 3e) nor in NPTX2 serum levels over time across all pwSMA (Figure 3d).

The CNTN5 serum but not CSF level was higher in controls than in pwSMA at baseline (z = 2.219, *p* = 0.026, *n* = 15). There were no changes in any other CNTN5 and ANTXR1 levels over time or differences between controls and SMA Improvers and SMA Non-Improvers. Levels of CNTN5 and ANTXR1 did not correlate with motor scores.

### 3.3. NPTX2 Immunostaining of Murine Spinal Cord Tissue Slices

The NPTX2 immunostaining fluorescence intensity was higher in SMA mice compared to wt mice at P42 (z = 2.366, *p* = 0.018, *n* = 7; three animals per condition with two or three slices per animal; wt mice not included in the analysis as they serve as reference values). NPTX2 intensities did not differ between SMA and wt mice at P20, P70, or P>400. The motoneuron count was significantly lower in SMA mice compared to wt mice at P42 (z = −2.375, *p* = 0.018, *n* = 7), P70 (z = −2.366, *p* = 0.018, *n* = 7), and P>400 (z = −2.677, *p* = 0.007, *n* = 9) but not at P20, indicating motoneuronal loss had begun by P42 (Figure 5).

## 4. Discussion

The level of NPTX2, a protein of the neuronal pentraxin family involved in synaptic function and plasticity, decreased in CSF in adult pwSMA in the first 14 months of treatment with nusinersen independent of motor improvement. The change in NPTX2 serum levels over 14 months of nusinersen treatment correlated with the change in HFMSE during this period. This demonstrates the potential of NPTX2 as a CSF biomarker reflecting a biological response to nusinersen therapy and as a serum biomarker associated with clinical response patterns, with opposing trajectories observed in SMA Improvers and Non-Improvers. The upregulation of NPTX2 in the spinal cord with the onset of motor neuron loss in the SMA mouse model supports the biological significance of the CSF and serum findings.

NPTX2 is expressed in the brain and spinal cord but also in peripheral tissues, such as the Langerhans islets and the adrenal medulla, and is involved in the assembly of excitatory synapses [23]. Notably, abnormalities in the Langerhans islets have been reported in SMA mouse models and in children with severe infantile form of the disease, linking this peripheral expression site of NPTX2 to pathological processes in SMA [24,25].

The presynaptic expression and release of NPTX2 is upregulated by synaptic activity and also by brain-derived neurotrophic factor (BDNF) [26]. After being released, NPTX2 can accumulate at excitatory synapses on interneurons within perineuronal nets [27]. Its function there has not yet been fully clarified. However, building a complex with Neuronal Pentraxin 1 (NPTX1), it binds and aggregates α-amino-3-hydroxy-5-methyl-4-isoxazolepropionic acid (AMPA) receptors [28,29]. It thus can modulate synaptic plasticity and in particular the transmission of excitatory signals via glutamatergic synapses and appears to stabilize AMPA receptors and thus increase their activity [30]. Under certain stress conditions, e.g., with upregulation of tumor necrosis factor-α converting enzyme, Neuronal Pentraxin Receptor (NPTXR) clusters with NPTX2 and AMPA receptors, leading to the internalization of this complex [31]. Investigations in an ischemic stroke rat model suggested that the induced clustering of AMPA receptors by NPTX2 leads to their internalization, resulting in reduced response to glutamate, reflecting a rescue mechanism. Thus, upregulation of NPTX2 may protect neurons from glutamate excitotoxicity in environments with excess of glutamate in the synaptic cleft as in ischemic stroke [32], but also in adult SMA. Glutamate excitotoxicity has been shown to play a critical role in motor neuron loss in adult SMA. One of our previous studies demonstrated that the excitatory amino acid transporter 1 (EAAT1), a glutamate transporter expressed in astrocytes, was downregulated in the spinal cord of a late-onset SMA mouse model and in cultured SMN-deficient astrocytes [33]. Decreased EAAT1 expression was associated with increased glutamate levels in the spinal cord [33]. Another group validated the reduction in EAAT1 in lumbar spinal cord tissue of individuals with SMA and in human iPSC-derived astrocytes [34]. The activation of spinal astrocytes and the decreased expression of EAAT1 with increased glutamate levels even preceded the loss of spinal motor neurons in the late-onset SMA mouse model [22]. This early astrocytic pathomechanism of SMA may involve NPTX2 via its influence on glutamatergic synapses.

Considering the biological function of NPTX2, the decrease in NPTX2 CSF level during nusinersen therapy might reflect a downregulated rescue mechanism as a consequence of SMN restoring therapy. However, an upregulation of NPTX2 at baseline could not be demonstrated in pwSMA compared to controls. The divergent dynamics observed in CSF and serum remain unclear, and may, at least in part, be attributable to the limited sample size as well as to the cross-reactivity and limited sensitivity of the ELISA, particularly in the context of low protein concentrations in CSF. Furthermore, the dynamics of NPTX2 should be interpreted in light of the overall low disease dynamics in our cohort, predominantly consisting of SMA type 3, characterized by long disease duration and a slow rate of clinical progression. This may have limited the detectability of treatment-related biomarker changes, especially within a follow-up period of 14 months. Whether more pronounced NPTX2 dynamics occur in more rapidly progressive phenotypes, such as SMA types 1 and 2, remains to be explored in future studies. Moreover, individuals with SMA type 3 often exhibit substantial interindividual variability in disease duration, age, and motor function, which may further dilute potential group effects in small samples. One advantage of including individuals with SMA type 3 is that this group can be reasonably well characterized using the HFMSE and RULM scores, as the known floor and ceiling effects of these functional measures are relatively limited in this phenotype [35]

Serum NPTX2 levels have been analyzed in previous studies and have been found to be altered in individuals with acute psychotic episodes in schizophrenia [36] and to be associated with cognitive function in patients with dementia [37]. Also, CSF NPTX2 levels as measured by ELISA, have been found to be reduced in individuals with Alzheimer’s Dementia compared to age-matched controls. NPTX2 concentration was even shown to correlate with cognitive performance and hippocampal volume in this patient group [20]. Postmortem analyses of human cortex tissue from this study revealed a decreased NPTX2 level, supporting the biological significance of the CSF findings [20].

Our small-scale analyses of spinal cord sections in the established late-onset SMA mouse model [22] revealed an upregulation of NPTX2 in the ventral horn of the spinal cord on day 42, when motor neuron loss had already begun, aligning with our human CSF and serum findings. While these analyses do not aim to elucidate pathomechanisms, they underscore the relevance of NPTX2 concentrations in human CSF and serum as potential biomarkers.

Our findings suggest that NPTX2 plays a role in the pathomechanism of adult SMA. As a biomarker in CSF, it may provide biological-mechanistic insights into the therapeutic response to SMN-restoring therapy with nusinersen. Measured in serum, it could be useful as a biomarker to objectify the clinical-functional treatment response, though our results do not support that it allows for its prediction.

Multi-omics analyses, performed by mass spectrometric analyses, have become an essential tool in understanding underlying molecular mechanisms and identifying biomarkers especially of rare diseases like SMA [38,39]. Proteomic studies, for example, have demonstrated that CSF proteins involved in axonogenesis, as well as bioenergetic and inflammatory pathways, exhibit differential expression in pwSMA compared to controls, and also before and after nusinersen treatment [40,41]. Notably, these effects were evident even in small sample sizes, with as few as 10 to 13 pwSMA [39,42]. However, the technique faces challenges, including difficulties in detecting low-abundance proteins, the complexity of data analysis, and technical limitations such as sample degradation and incomplete coverage of the proteome. Furthermore, techniques are not standardized, making the results of different studies often poorly comparable.

CNTN5 is a protein of the contactin family, playing a role in cell adhesion and neuronal development. It is primarily expressed in the central nervous system and is involved in synapse formation and function. It has been studied as a biomarker in autoimmune nodopathies [43]. ANTXR1 is a membrane protein which is primarily expressed on the surface of cells, particularly endothelial cells and macrophages. Anthrax toxins receptors were shown to be expressed in the nervous system, especially in sensory neurons [44]. While significant differences in CNTN5 and ANTXR1 levels between improvers and non-improvers were observed in the proteomic analysis, we could not verify this in the quantitative measurement using ELISA in serum or CSF. The lack of concordance between the results of the proteomic analysis and the quantification via ELISA may be due to the different sensitivity and specificity of the tests, a different type of sample preparation and processing for the proteome analysis, different calibration strategies and reference standards and lower resolving power with, e.g., interferences in the proteome analysis. In view of the increasing use of new explorative methods such as proteome analysis, our data also demonstrate that the results should be interpreted with caution and should be confirmed by quantitative analyses.

Limitations of our study include the sample size with some missing data for later time points. The relatively small sample size increases the risk of both false negative and false positive findings. A statistically well-powered study would require substantially larger cohorts. The findings should therefore be considered exploratory and warrant validation in larger cohorts. The sample selection, characterized by very heterogeneous treatment responses as indicated by changes in HFMSE scores, may potentially confound the correlational statistics with the biomarkers and is not optimally representative of a real-world SMA cohort. Limitations of proteomic analysis primarily include the complexity and dynamics of the proteome, with proteins existing in various isoforms, complicating identification and quantification, as well as the general detection threshold and quantification accuracy. The limitations of the ELISA assay include potential cross-reactivity and limited sensitivity.

## 5. Conclusions

In this study, we were able to identify NPTX2 through a simple ELISA analysis in a small cohort as a promising biomarker to objectify the treatment response of adults with SMA to nusinersen. These findings should be confirmed in a larger cohort to validate the results within a more robust large-scale framework. Furthermore, our results imply a significant pathophysiological role of NPTX2 in late-onset SMA, demanding further investigation. It would be particularly interesting to closer investigate the interactions between NPTX2, NPTX1, and NPTXR, considering their formation of complexes and interactions in their expression patterns [30].

## Figures and Tables

**Figure 1 biomedicines-13-01821-f001:**
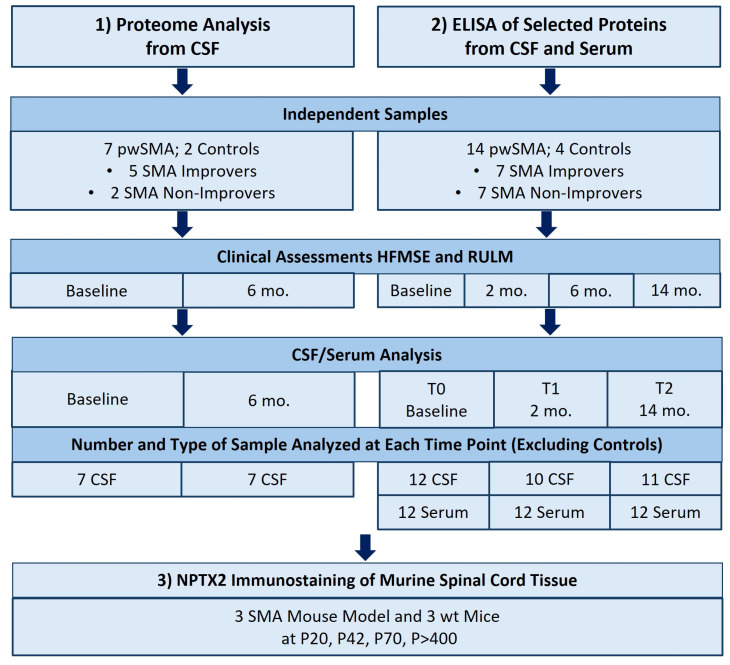
Overview of the study design. Control samples were only available from a single time point. pwSMA = adult individuals with 5q-SMA; SMA Improvers = increase ≥ 3 points in HMFSE after 6 months of treatment with nusinersen; SMA Non-Improvers = increase of < 3 points in HFMSE after 6 months of treatment with nusinersen; HFMSE = Hammersmith Functional Motor Scale Expanded; RULM = Revised Upper Limb Module; mo. = months; wt = wild type; P20, P40, P72, and P>400 refer to postnatal days, with P20 indicating postnatal day 20.

**Figure 2 biomedicines-13-01821-f002:**
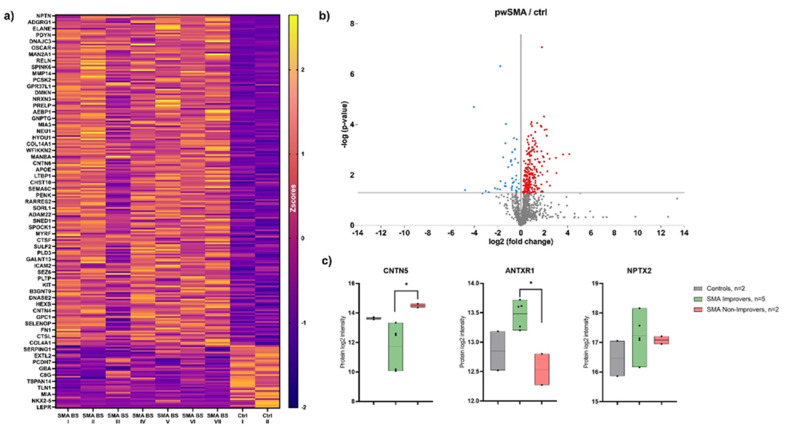
Results of the proteomic analysis with (**a**) Heatmap of differentially expressed proteins between pwSMA and controls (Ctrl). The color scale on the right represents Z-score values, with yellow shades indicating increased expression levels and purple shades indicating decreased expression levels. SMA BS (Baseline) I to V represent SMA Improvers (HFMSE improvement ≥ 3 points after 6 months of nusinersen therapy), while SMA BS VI and VII represent SMA Non-Improvers (HFMSE improvement < 3 points). Each row corresponds to a protein and each column represents a sample. The visual cluster structure of the heatmap reveals differences in protein expression between individuals with SMA and controls; (**b**) Volcano Plot of differentially expressed proteins between pwSMA and controls (ctrl). The logarithmic change in expression levels (log2 fold change) is represented on the X-axis, while the statistical significance is shown as the negative decadic logarithm of the *p*-value (−log10(*p*-value)) on the Y-axis. Each point represents a different protein, with color-coded points indicating statistically significant differences (*p* < 0.05) between pwSMA and controls. Red points indicate a positive change in protein expression compared to controls, while blue points represent a negative change; (**c**) Results of proteomic profiling from CSF with baseline levels of CNTN5, ANTXR1, and NPTX2 in controls and SMA Improvers and Non-Improvers. Significant differences (*p* <0.05) are marked with an asterix (*). CNTN5 levels are decreased in SMA Improvers compared to Non-Improvers, with no significant difference compared to controls. ANTXR1 is elevated in SMA Improvers compared to Non-Improvers, again without a significant difference compared to controls. For NPTX2, no significant group differences were observed.

**Figure 3 biomedicines-13-01821-f003:**
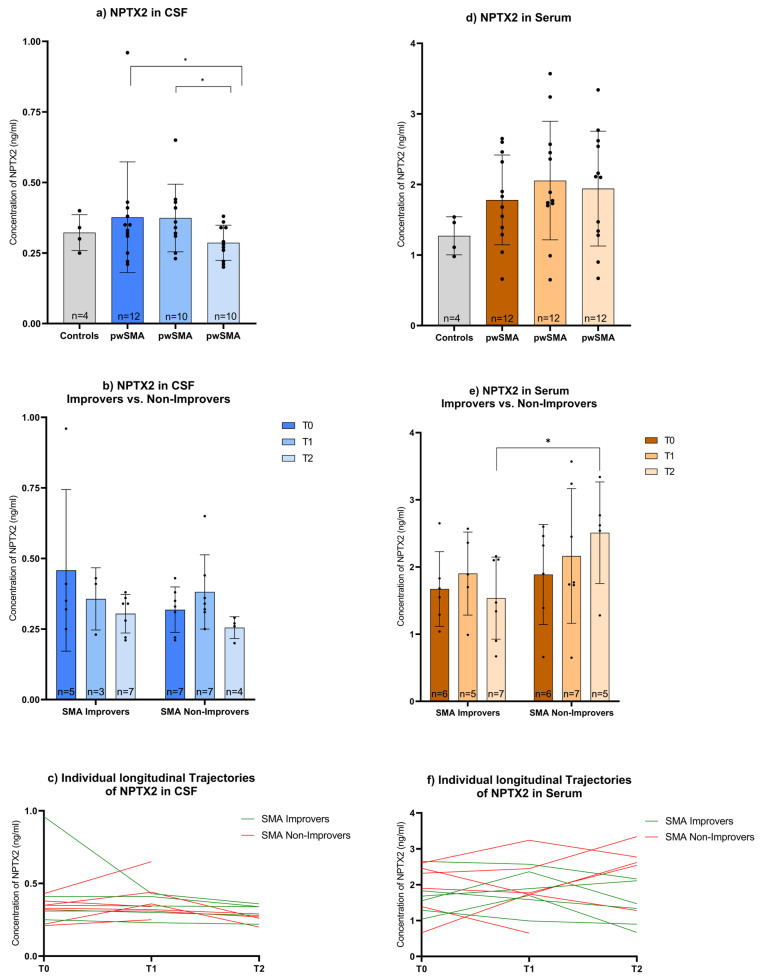
(**a**) CSF and (**d**) Serum levels of NPTX2 in controls and pwSMA at T0 (baseline), T1 (2 months after start of treatment nusinersen), and T2 (14 months after start of treatment with nusinersen) across all pwSMA shown in a bar plot with individual data points and error bars representing the standard deviation. The bar height represents the mean value. Across the entire pwSMA cohort, CSF NPTX2 levels were lower at T2 compared to T0 and lower at T2 compared to T1. No significant differences were observed in serum levels across time points. (**b**) CSF and (**e**) Serum levels of NPTX2 at T0, T1, and T2 in SMA Improvers (improvement in HFMSE ≥ 3 points after 6 months of nusinersen therapy) vs. SMA Non-improvers (improvement in HFMSE < 3 points after 6 months of nusinersen therapy). Significant differences (*p* < 0.05) are marked by an asterisk *. Note that the Y-axis scaling differs between serum and CSF values due to a different range of protein concentration. At T2, serum NPTX2 levels were lower in SMA Improvers compared to Non-Improvers. No significant differences in CSF NPTX2 levels were observed between SMA Improvers and Non-Improvers at the respective time points. Individual longitudinal trajectories of NPTX2 (**c**) CSF and (**f**) serum levels are shown in a line graph across all pwSMA, with SMA Improvers highlighted in green and SMA Non-Improvers in red.

**Figure 4 biomedicines-13-01821-f004:**
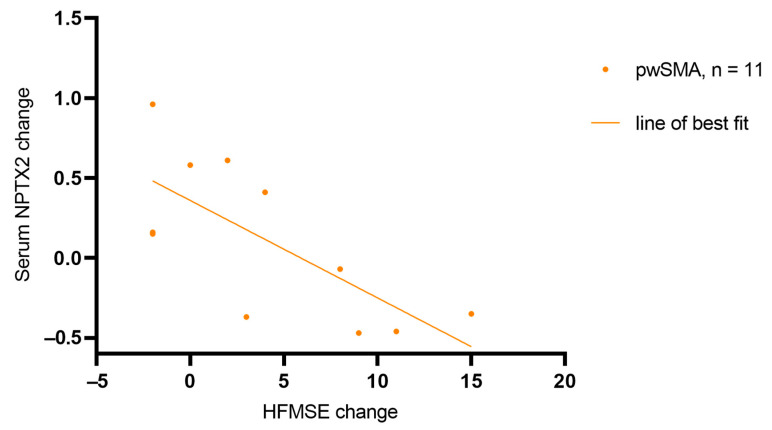
Correlation of the change in NPTX2 serum concentration and HFMSE score from T0 (baseline) to T2 (14 months after treatment initiation with nusinersen) (r = −0.688, *p* = 0.019, *n* = 11). The points represent individual pwSMA. The line of best fit was calculated using simple linear regression, R^2^ = 0.52, *p* = 0.01.

**Figure 5 biomedicines-13-01821-f005:**
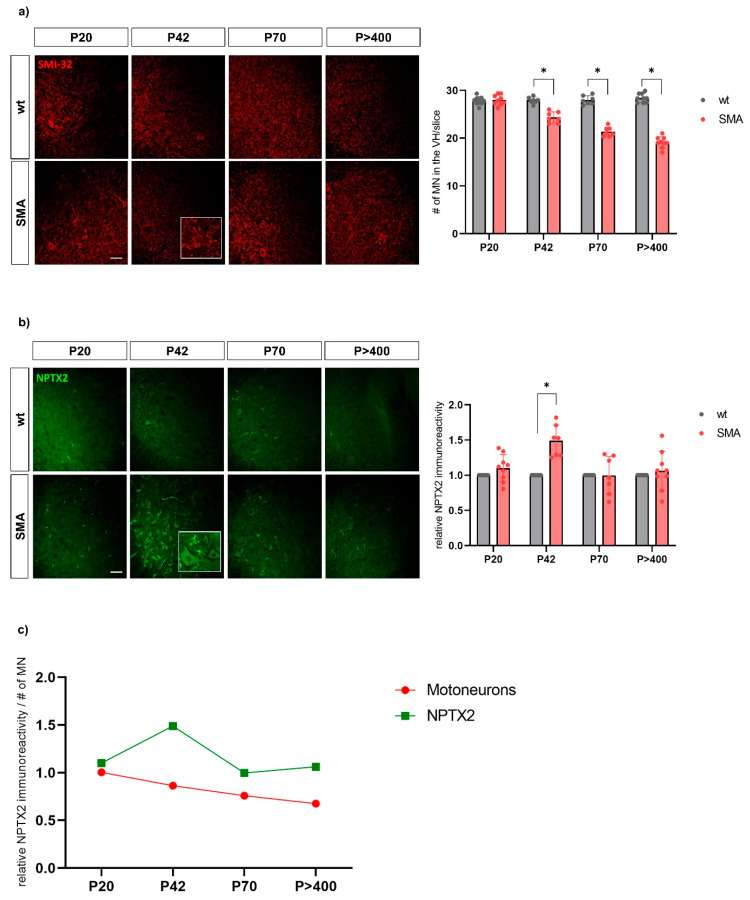
Immunostaining of (**a**) spinal motoneurons (SMI-32, red) and (**b**) NPTX2 (green) in the ventral horn of lumbar spinal cord slices from wild type (wt) and SMA mice (SMA) at P20 (postnatal day 20), P42 and P70 and P>400. The SMA values normalized to the wt values for the immunoreactivity of NPTX2 and the absolute number of motoneurons at the different time points are shown on the right (*n* = 9 for P20 and P>400; *n* = 7 for P42 and P70; three SMA mice per condition with two or three slices per animal). Scale bar: 20 µm. From P42 onwards, the number of motor neurons is reduced in SMA mice compared to wt mice. At P42–but not at any other time point–NPTX2 immunofluorescence is increased in SMA compared to wt mice. Significant differences (*p* < 0.05) are marked by an asterisk *. (**c**) Dynamics of wt-normalized values for motoneuron count and NPTX2 immunoreactivity in SMA mice across the different time points. In SMA mice, the number of motor neurons continues to decline progressively from P42 onwards, while NPTX2 immunofluorescence peaks at P42.

**Table 1 biomedicines-13-01821-t001:** Demographic and clinical characteristics of all pwSMA included in ELISA (*n* = 14) and proteome (*n* = 7) analysis.

Sex	Male (%)	Female (%)	
Proteome analysis	5 (71)	2 (29)	
ELISA Analysis	10 (71)	4 (29)	
**SMA type**	**2**	**3**	
Proteome analysis	2 (29)	5 (71)	
ELISA Analysis	1 (7)	13 (93)	
***SMN2* copy number**	**2 (%)**	**3 (%)**	**4 (%)**
Proteome analysis	0 (0)	3 (43)	4 (57)
ELISA Analysis	1 (7)	6 (43)	7 (50)
**Clinical classification**	**Non-Sitter**	**Sitter**	**Walker**
Proteome analysis	0 (0)	4 (57)	3 (43)
ELISA Analysis	4 (29)	1 (7)	9 (64)
**Spondylodesis**	**Yes (%)**	**No (%)**	
Proteome analysis	1 (14)	6 (86)	
ELISA Analysis	2 (14)	12 (86)	
**Non-invasive ventilation**	**Yes (%)**	**No (%)**	
Proteome analysis	2 (29)	5 (71)	
ELISA Analysis	3 (21)	11 (79)	
**Age**	**Mean (sd ^a^)**	**Minimum**	**Maximum**
Proteome analysis	40.00 (14.28)	26	65
ELISA Analysis	37.71 (12.86)	22	62
**Disease duration in years**	**Mean (sd)**	**Minimum**	**Maximum**
Proteome analysis	35.01 (16.22)	14	63
ELISA Analysis	29.08 (14.34)	12	61
**Baseline HFMSE score**	**Mean (sd)**	**Minimum**	**Maximum**
Proteome analysis	28.00 (22.06)	0	57
ELISA Analysis	32.50 (21.38)	1	62
**Baseline RULM score**	**Mean (sd)**	**Minimum**	**Maximum**
Proteome analysis	25.00 (13.90)	0	37
ELISA Analysis	29.64 (10.49)	7	37

^a^ standard deviation

## Data Availability

The protein ratios between the groups obtained from the proteomic analysis are available as a Appendix A. The mass spectrometry proteomics data have been deposited to the Proteo-meXchange Consortium via the PRIDE partner repository with the dataset identifier PXD065345. The other datasets generated and analyzed during the current study are not publicly available due to the inclusion of clinical patient data that cannot be disclosed for data protection and privacy reasons but are available from the corresponding author on reasonable request.

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
