# Peer review of "Neuronal Pentraxin 2 as a Potential Biomarker for Nusinersen Therapy Response in Adults with Spinal Muscular Atrophy: A Pilot Study"

_biomedicines, 2025, doi:10.3390/biomedicines13081821_

Round 1
Reviewer 1 Report
Comments and Suggestions for Authors
Review of the Manuscript: “Neuronal Pentraxin 2 as a Potential Biomarker for Nusinersen Therapy Response in Adults with Spinal Muscular Atrophy: A Pilot Study
This manuscript presents a case-control study utilizing proteomics to identify potential bio-markers that may help predict or objectively assess treatment response in patients with spinal muscular atrophy (SMA) undergoing Nusinersen therapy.
Overall, the study is well-written and contributes valuable insights into the biological response to Nusinersen therapy. It holds promise for future efforts to stratify responders from non-responders. The manuscript has been reviewed in accordance with the STROBE guidelines for case-control studies. Below are minor and major suggestions for improvement:
Minor Revisions
- Include a clear description of the study design in the Methods section.
- Provide details on the selection criteria for healthy controls.
- Explain the rationale for the study size and include a power calculation.
Major Revision
- Discuss the potential impact of the study population being predominantly composed of SMA type 3 patients. Specifically, elaborate on the statement: “Furthermore, the dynamics of NPTX2 should be interpreted in light of the overall low dis-ease dynamics in our cohort, predominantly consisting of SMA type 3, characterized by long disease duration and a slow rate of clinical progression.”
- Clarify why CNTN5, ANTXR1, and NPTX2 are not represented in the heat map (Figure 2a).
- In the Results section, the sentence: “In serum, the NPTX2 concentration showed no differences between pwSMA and controls (Figure 3c), nor between SMA Improvers and SMA Non-Improvers at T0 or T1 (Figure 3d),” would be clearer if moved further down, after presenting the following finding: "The NPTX2 serum level was higher at T2 in SMA Non-Improvers than in SMA Improvers (z=-2.030, p=0.48, n=12) (Figure 3d). It increased in all 5 SMA Non-Improvers from T0 to T2 and decreased in 5 of 6 SMA Improvers from T0 to T2, thus NPTX2 serum levels did not change over time across all pwSMA (Figure 3c) and the NPTX2 concentration showed no differences between pwSMA and controls (Figure 3c), nor between SMA Improvers and SMA Non-Improvers at T0 or T1(Figure 3d).”
- Clarify whether the reported negative correlation between serum NPTX2 changes and HFMSE changes was observed in the pooled group of improvers and non-improvers, or with-in a specific subgroup.
- The statement:
“This demonstrates the potential of NPTX2 as a CSF biomarker indicating a biological re-sponse to Nusinersen therapy and a serum biomarker displaying clinical treatment response.”
should be revised to reflect that this conclusion is primarily supported by findings in the sub-group of clinical improvers.
Author Response
Reply to Reviews
We would like to thank the reviewers for their valuable comments and suggestions, which we have used to improve our manuscript for this revision. We have carefully addressed each of their suggestions and criticisms. All authors have read the revised form and agree to this form. Below please find our point-by-point reply.
Response to Reviewer #1:
Point 1: Include a clear description of the study design in the Methods section.
Response 1: We have revised Section 2.1 (Study Design) and provided a more detailed and comprehensive description of the study design and the three main analyses. Figure 1 additionally provides a schematic overview of the study design.
Point 2: Provide details on the selection criteria for healthy controls.
Response 2: Additional information regarding this was added to Section 2.1 (Study Design), paragraph 2.
Point 3: Explain the rationale for the study size and include a power calculation.
Response 3: This study was designed as a pilot investigation and therefore included a relatively small sample size. Power analysis indicated that a minimum of 20 to 25 individuals per group would be required to reliably detect longitudinal differences in CSF and serum biomarker levels between responders and non-responders. As the current study included smaller group sizes - also due to feasibility constraints - the findings should be considered exploratory and require confirmation in larger, adequately powered cohorts. We have addressed this point by adding a corresponding statement to the limitations section (final paragraph of the Discussion section).
Point 4: Discuss the potential impact of the study population being predominantly composed of SMA type 3 patients. Specifically, elaborate on the statement: “Furthermore, the dynamics of NPTX2 should be interpreted in light of the overall low dis-ease dynamics in our cohort, predominantly consisting of SMA type 3, characterized by long disease duration and a slow rate of clinical progression.”
Response 4: We appreciate the reviewer’s comment and agree that the composition of the study population - primarily individuals with SMA type 3 - has important implications for the interpretation of our findings. SMA type 3 is typically associated with a later onset, slower disease progression, and overall lower clinical and potentially biological activity compared to types 1 and 2. As a result, changes in biomarkers such as NPTX2 may occur more subtly or more slowly over time in this subgroup. This relatively stable disease course may limit the detectability of dynamic biomarker responses, especially within a follow-up period of 14 months. Whether more pronounced NPTX2 dynamics occur in more rapidly progressive phenotypes, such as SMA types 1 and 2, remains to be explored in future studies. Moreover, individuals with SMA type 3 often exhibit substantial interindividual variability in disease duration, age, and motor function, which may further dilute potential group effects in small samples. One advantage of including individuals with SMA type 3 is that this group can be reasonably well characterized using the HFMSE and RULM scores, as the known floor and ceiling effects of these functional measures are relatively limited in this phenotype. We have added this point accordingly to the Discussion, paragraph 4.
Point 5: Clarify why CNTN5, ANTXR1, and NPTX2 are not represented in the heat map (Figure 2a).
Response 5: The heatmap displays differentially expressed proteins between pwSMA and controls, not between SMA Improvers and SMA Non-Improvers. Although the main focus of this study is the comparison between SMA Improvers and Non-Improvers, presenting a heatmap of differentially expressed proteins between pwSMA and controls at baseline was thought to be useful to provide a broader context. It offers an overview of the general proteomic alterations associated with SMA and helps to illustrate the distribution and structure of the dataset before focusing on treatment-related subgroup analyses.
CNTN5, ANTXR1, and NPTX2 were not found to be differentially regulated (p-value < 0.05 and average fold change > 1.5) between pwSMA and controls in the proteomic analysis (see Results section) and therefore do not appear in the heatmap. In Figure 2c, the results of the group comparison between SMA Improvers and SMA Non-Improvers for CNTN5, ANTXR1, and NPTX2 are visualized.
Point 6: In the Results section, the sentence: “In serum, the NPTX2 concentration showed no differences between pwSMA and controls (Figure 3c), nor between SMA Improvers and SMA Non-Improvers at T0 or T1 (Figure 3d),” would be clearer if moved further down, after presenting the following finding: "The NPTX2 serum level was higher at T2 in SMA Non-Improvers than in SMA Improvers (z=-2.030, p=0.48, n=12) (Figure 3d). It increased in all 5 SMA Non-Improvers from T0 to T2 and decreased in 5 of 6 SMA Improvers from T0 to T2, thus NPTX2 serum levels did not change over time across all pwSMA (Figure 3c) and the NPTX2 concentration showed no differences between pwSMA and controls (Figure 3c), nor between SMA Improvers and SMA Non-Improvers at T0 or T1 (Figure 3d).”
Response 6: We have revised this in Results Section 3.2. Only the first part, “In serum, the NPTX2 concentration showed no differences between pwSMA and controls (Figure 3c),” was kept at the beginning to maintain consistency in presenting the SMA vs. control comparison first for both CSF and serum.
Point 7: Clarify whether the reported negative correlation between serum NPTX2 changes and HFMSE changes was observed in the pooled group of improvers and non-improvers, or within a specific subgroup.
Response 7: It was observed in the pooled group. We specified this in Section 3.2.
Point 8: The statement “This demonstrates the potential of NPTX2 as a CSF biomarker indicating a biological response to Nusinersen therapy and a serum biomarker displaying clinical treatment response” should be revised to reflect that this conclusion is primarily supported by findings in the subgroup of clinical improvers.
Reponse 8: We appreciate the reviewer’s comment. However, we would like to clarify that the serum NPTX2 dynamics were observed in both SMA Improvers and Non-Improvers, in opposite directions. The significant correlation between NPTX2 change and HFMSE change was calculated across all pwSMA, not limited to a subgroup. We clarified this as addressed in Response 7. Therefore, the interpretation that the serum findings primarily reflect the subgroup of clinical improvers may be too narrow. Instead, our data suggest that NPTX2 dynamics in serum may reflect treatment response patterns across the entire cohort, with inverse trends depending on clinical improvement. To address this point and improve clarity, we have revised the sentence in the discussion to “This demonstrates the potential of NPTX2 as a CSF biomarker reflecting a biological response to nusinersen therapy and as a serum biomarker associated with clinical response patterns, with opposing trajectories observed in SMA Improvers and Non-Improvers”. Accordingly, this part of the Results section in the abstract was slightly revised to emphasize the correlation between HFMSE and the change in NPTX2 serum levels rather than the difference in NPTX2 levels at T2 between Improvers and Non-Improvers.
non-improvers).
Additional Changes:
- Gene names that were not yet italicized have been changed to italics.
- As we noticed a somewhat ambiguous formulation in the final paragraph of Section 2.8, we revised the wording for clarity without making any changes to the content.
Reviewer 2 Report
Comments and Suggestions for Authors
In this work, the authors present the results of a pilot study to identify biomarkers associated with treatment response in SMA patients. The paper is interesting, and the topic is relevant. However, some issues should be addressed before recommending it for publication:
- As their first step, the authors perform a proteomic analysis of 7 SMA patients and two controls. Why was this analysis so unbalanced? Including only two controls could have biased their results. The same applies to balancing improvers (5 samples) and non-improvers (2 samples). I understand the limitations in sample size for a pilot study. Still, the authors should provide arguments for not having balanced data (i.e., three controls, three improvers, and three non-improvers).
- As the second step, the authors perform ELISA and, again, do it with an unbalanced sample.
- The comparison of T0, T1, and T2 biomarker values should be performed using methods for longitudinal data analysis. The authors have performed multiple comparisons using the Wilcoxon signed rank test, so the type I error will be inflated.
- FIgures of results should reflect the longitudinal nature of the data. Points from the same patients should be joined together with a line, so that readers can see individual trends.
Author Response
Reply to Reviews
We would like to thank the reviewers for their valuable comments and suggestions, which we have used to improve our manuscript for this revision. We have carefully addressed each of their suggestions and criticisms. All authors have read the revised form and agree to this form. Below please find our point-by-point reply.
Response to Reviewer #2:
Point 1: As their first step, the authors perform a proteomic analysis of 7 SMA patients and two controls. Why was this analysis so unbalanced? Including only two controls could have biased their results. The same applies to balancing improvers (5 samples) and non-improvers (2 samples). I understand the limitations in sample size for a pilot study. Still, the authors should provide arguments for not having balanced data (i.e., three controls, three improvers, and three non-improvers).
Response 1: We thank the reviewer for this comment. The proteomic analysis was intended as an exploratory screening to identify candidate biomarkers, not for statistical group comparison. Moreover, we expected that the higher number of Improvers might increase the likelihood of detecting potential therapy-related proteomic changes, as this subgroup was anticipated to show greater biological dynamics in correlation with more pronounced clinical change. Proteomic analyses were considerably more expensive at the time, and baseline samples from SMA patients are limited. Therefore, sample selection was based on availability, technical suitability (e.g., volume, quality) and clinical dynamics rather than on balanced group sizes. We added an explanation in the Methods Section 2.3.
Point 2: As the second step, the authors perform ELISA and, again, do it with an unbalanced sample.
Response 2: A total of 14 pwSMA were included in the ELISA analysis, with 7 classified as Improvers and 7 as Non-Improvers. This is described in more detail in the Methods section 2.4 and illustrated in Figure 1. We acknowledge that the number of controls (n=4) is comparatively small. However, since the primary focus of this analysis was on differences in clinical treatment response among pwSMA rather than on comparisons between patients and controls, the control group was not expanded further.
Point 3: The comparison of T0, T1, and T2 biomarker values should be performed using methods for longitudinal data analysis. The authors have performed multiple comparisons using the Wilcoxon signed rank test, so the type I error will be inflated.
Response 3: Thank you for the comment. For the comparison of NPTX2 levels across T0, T1, and T2, a Friedman test was now applied for the longitudinal analysis, as the data were not all normally distributed. This was followed by a post hoc analysis using the Wilcoxon signed-rank test with Bonferroni correction of the significance level. The corresponding changes have been made in Methods Section 2.8 and Results Section 3.2.
Point 4: Figures of results should reflect the longitudinal nature of the data. Points from the same patients should be joined together with a line, so that readers can see individual trends.
Response 4: We added new parts to Figure 3, showing the longitudinal course of NPTX2 serum and CSF levels in SMA Improvers and Non-Improvers.
Additional Changes:
- Gene names that were not yet italicized have been changed to italics.
- As we noticed a somewhat ambiguous formulation in the final paragraph of Section 2.8, we revised the wording for clarity without making any changes to the content.
Round 2
Reviewer 2 Report
Comments and Suggestions for Authors
The authors have addressed all my previous comments.